# Postmortem Electrical Conductivity Changes of *Dicentrarchus labrax* Skeletal Muscle: Root Mean Square (RMS) Parameter in Estimating Time since Death

**DOI:** 10.3390/ani12091062

**Published:** 2022-04-20

**Authors:** Jessica Maria Abbate, Gabriele Grifò, Fabiano Capparucci, Francesca Arfuso, Serena Savoca, Luca Cicero, Giancarlo Consolo, Giovanni Lanteri

**Affiliations:** 1Department of Veterinary Sciences, University of Messina, Polo Universitario Annunziata, 98168 Messina, Italy; jabbate@unime.it (J.M.A.); farfuso@unime.it (F.A.); 2Department of Mathematical, Computer, Physical and Earth Sciences, University of Messina, Polo Universitario Papardo, 98166 Messina, Italy; gabriele.grifo@unime.it (G.G.); giancarlo.consolo@unime.it (G.C.); 3Department of Chemical, Biological, Pharmaceutical and Environmental Sciences, University of Messina, Polo Universitario Papardo, 98166 Messina, Italy; fcapparucci@unime.it (F.C.); glanteri@unime.it (G.L.); 4Department of Biomedical, Dental Sciences and Morphological and Functional Images, University of Messina, 98125 Messina, Italy; ssavoca@unime.it; 5National Research Council, Institute for Marine Biological Resources and Biotechnology (IRBIM), 98122 Messina, Italy; 6Experimental Zooprophylactic Institute of Sicily “A. Mirri” (IZS), 98129 Palermo, Italy

**Keywords:** postmortem interval, PMI, electrical conductivity, RMS, skeletal muscle, *Dicentrarchus labrax*, teleost

## Abstract

**Simple Summary:**

The estimation of postmortem interval (PMI) still poses a major challenge for pathologists worldwide, making the search for new and more accurate technologies to assist in PMI estimation worthy of growing scientific interest. This study aimed to explore for the first time the use of an oscilloscope coupled with a signal generator, as innovative technology, to evaluate changes in the electrical conductivity of skeletal muscle of sea bass specimens during the early postmortem interval, to find an accurate, quantitative parameter useful in PMI estimation. The use of the oscilloscope, especially for the RMS measured parameter, has been shown here as a promising technology for studying dielectric muscle properties during the early postmortem interval, with the advantage of being a rapid, non-destructive, and inexpensive method.

**Abstract:**

Electric impedance spectroscopy techniques have been widely employed to study basic biological processes, and recently explored to estimate postmortem interval (PMI). However, the most-relevant parameter to approximate PMI has not been recognized so far. This study investigated electrical conductivity changes in muscle of 18 sea bass specimens, maintained at different room temperatures (15.0 °C; 20.0 °C; 25.0 °C), during a 24 h postmortem period using an oscilloscope coupled with a signal generator, as innovative technology. The root mean square (RMS) was selected among all measured parameters, and recorded every 15 min for 24 h after death. The RMS(t) time series for each animal were collected and statistically analyzed using MATLAB^®^. A similar trend in RMS values was observed in all animals over the 24 h study period. After a short period, during which the RMS signal decreased, an increasing trend of the signal was recorded for all fish until it reached a peak. Subsequently, the RMS value gradually decreased over time. A strong linear correlation was observed among the time series, confirming that the above time-behaviour holds for all animals. The time at which maximum value is reached strongly depended on the room temperature during the experiments, ranging from 6 h in fish kept at 25.0 °C to 14 h in animals kept at 15.0 °C. The use of the oscilloscope has proven to be a promising technology in the study of electrical muscle properties during the early postmortem interval, with the advantage of being a fast, non-destructive, and inexpensive method, although more studies will be needed to validate this technology before moving to real-time field investigations.

## 1. Introduction

The estimation of the postmortem interval (PMI), known as time since death, is a critical and important issue in some animal death investigations, and, to date, it still represents a major challenge for veterinary pathologists globally [1,2]. An accurate PMI estimation may be crucial in identifying the cause and manner of death, distinguishing between ante- and perimortem trauma, and, in animal forensic cases, it represents an issue of critical importance when the duration of abuse, neglect, or insurance fraud must be established [1,2]. While a critical investigative tool, accurate PMI estimation is complex, becoming increasingly so as the PMI prolongs [2,3,4].

Following death, several progressive and irreversible biochemical and physical modifications occur in the body, leading to the complete disintegration of the carcass [1]. PMI estimation is strongly dependent upon knowledge of these postmortem changes, as well as their expected onset and progression on the animal carcass [1,2,3,4]. In particular, the cooling rate of the body (e.g., algor mortis), livor mortis, and the onset and resolution of the rigor mortis phenomenon are the most commonly addressed postmortem changes [5,6,7,8]. However, while physical changes usually occur in a predictable order, the rate and manner at which they develop are strongly influenced by several environmental conditions (i.e., external temperature; oxygen tension) and individual factors (i.e., animal species; body condition score; health status), thereby often invalidating the concrete application in the field practice of the data generated by experimental studies [1,2]. Chemical analyses of body fluids have been largely investigated in forensic science to find reliable and objective biomarkers to assist in PMI estimation [9], and, more recently, molecular methods have been explored to correlate the degradation of nucleic acids and tissue proteins with PMI [10,11,12,13,14]. However, although analytic methods have shown great promise in the PMI estimation, they are often expensive and of little practical value for field application [10,11,12,13,14]. Additionally, both conventional and innovative methods are not precise if used individually, and their joint use is mandatory for increasing overall accuracy [1,2]. Therefore, useful new technologies are necessary to estimate the PMI more accurately.

Investigations on the electrical proprieties of biological tissues have been of great interest for decades, with a wide range of practical applications. Electrical conductivity analysis is a rather simple, fast, non-destructive, and inexpensive method, and it has been widely employed in many fields, such as food science [15,16,17], clinical medicine [18,19], and forensic environmental science [20,21], and it was recently introduced to investigate cadaveric decomposition rate and PMI estimation [22,23,24]. Since electrical properties of tissues are strongly dependent on morphological and functional cells’ integrity and intra- and extracellular conductivities, electric impedance spectroscopy techniques have been employed to study basic biological processes, differentiating between organisms’ physiological and pathological status, to predict carcass decomposition and postmortem interval, although the most-relevant and accurate parameter to approximate PMI has not been recognized, to date [25,26,27,28,29,30]. Fish muscle exhibits significant changes in postmortem dielectric properties [31], and knowledge about these electrical changes may aid in time since death estimation. Therefore, the present study aims to investigate electrical conductivity changes in skeletal muscles from sea bass specimens (*Dicentrarchus labrax*), maintained under different laboratory conditions during a 24 h postmortem period, using an innovative technology which, to the best of our knowledge, has never been employed to date. Investigations on muscular bioelectric properties during the early postmortem interval in fish, as an animal model, would be useful in validating a rapid method for quantitative estimation of the time since death to be employed in forensic practice in the near future.

## 2. Materials and Methods

### 2.1. Fish and Experimental Procedures

A total of 18 sea bass (*Dicentrarchus labrax*) specimens, purchased from a fish farm (Acqua Azzurra S.p.A., Syracuse, Sicily, Italy), were used in this experimental study. Fish were reared in the Centre for Experimental Fish Pathology (Centro di Ittiopatologia Sperimentale della Sicilia–CISS) of the University of Messina (Italy), which has been accredited for the use and production of aquatic organisms for experimental research since 2006 (DM n 39 March 2006). Fish were fed twice a day with commercial pellets (Skretting, Italy) at 1.5% of body weight (BW), maintained in a 12 h light/dark cycle in a 500 L tank, with the following water-controlled conditions: 20–21 °C, salinity 35‰, pH 8.1, and dissolved oxygen (DO) 7 ppm. After 10 days of acclimatization, fish were individually euthanized (1 fish/day) with an overdose of anesthetic (2-phenoxyethanol > 0.7 mL/L), and after death, morphometric parameters (body weight; length) were recorded. Sex was determined at the end of the experimental procedures (24 h). Fish reared in the same tanks were scarified and used as animal sentinels to exclude infectious diseases, and representative portions of all organs were sampled, formalin-fixed, and routinely processed for histopathology. All experimental procedures were approved by the Ethical Committee of the Department of Veterinary Sciences of the University of Messina (approval no. 037/2019, 30 December 2019), and conducted according to the Italian (D.L. 26/2014) and European (2010/63/EU) regulations on use of animals for experiments.

### 2.2. Hematological Analyses and Electrolytes Measurements

From each fish, before being sacrificed, a blood sample was collected from the caudal vein using a 20 G × 1½ syringe, and collected in in two tube types: microtubes (Miniplast 0.6 mL, LP Italiana Spa, Milano) containing EDTA (ratio 1.26 mg/0.6 mL) as an anticoagulant agent, in order to assess the hematological profile, and falcon tubes without an anticoagulant agent in order to test the serum concentration of the electrolytes calcium, sodium, potassium and chloride. The hematological profile was assessed using the blood cell counter HeCo Vet C (SEAC, Florence, Italy), which was suitably modified by a specific software program designed for the hematological analysis of fish species [32,33,34]. Evaluation of the hemogram involved the determination of the red blood cell count (RBC), hematocrit (Hct), hemoglobin concentration (Hgb), white blood cell count (WBC), thrombocyte count (TC), mean corpuscular volume (MCV), mean corpuscular hemoglobin (MCH), and mean corpuscular hemoglobin concentration (MCHC).

Falcon tubes were centrifuged at room temperature at 1300× *g* for 10 min, and the obtained serum samples were stored at −20 °C until analysis. The serum concentrations of calcium, sodium, potassium, and chlorine were assessed using an automated clinical chemistry analyzer (Konelab 60I; Thermo Electron Corporation, Vantaa, Finland) by means of commercially available kits (Thermo Fisher Scientific Oy; Clinical Diagnostics Finland; Vantaa, Finland).

### 2.3. Experimental Animal Groups

Electrical conductivity in each sea bass was recorded for 24 h after death, under controlled laboratory conditions (30 ± 5% humidity; 12 h light/dark cycle). Room temperature (T_room_) was set at different values (15.0 °C; 20.0 °C; 25.0 °C) in the three different experiments. In particular, the electrical conductivity changes were assessed on fish maintained under the standard laboratory condition (20 °C; Group II: specimens #7–12), and then the effect of two different room temperatures (25 °C; Group I: specimens #1–6) (15 °C; Group III: specimens #13–18) was tested.

### 2.4. Experimental Setup and Measurements

A signal generator was employed to produce a square waveform having frequency 5 kHz, peak-to-peak value of 5 Volt, and mean value of 2.5 Volt (GwINSTEK AFG-2005, GOOD WELL INSTRUMENT Co., Ltd., No.7-1, Jhongsing Road, Tucheng Dist. New Taipei City, Taiwan DAkkS calibration). A timer (TV56, C.D.C. Elettromeccanica s.r.l., Brignano Gera d’Adda, Bergamo, Italia) was used, allowing the signal generator to produce a 1 s signal every 15 min. The signal (S) was applied to each specimen by placing several electrodes in the lateral skeletal muscles using 21 G needles as follows: the positive electrode (E1) was placed in the epaxialis muscle, behind the operculum, and the ground electrode (E2) was placed in the hypaxialis muscle, leaving a vertical distance of 2 cm between them. The first channel (C1) of the oscilloscope was connected to the same (E1)–(E2) needles to measure the input signal (GwINSTEK 2202E, GOOD WELL INSTRUMENT Co., Ltd., No.7-1, Jhongsing Road, Tucheng Dist. New Taipei City, Taiwan. Calibration procedure: GWS-Q3-2S-010). Then, several other electrodes were used to measure the output signal as follows: the first terminal of this probe (E3) was placed at a horizontal distance of 10 cm from electrode (E1), whereas the second one was connected to the common ground electrode (E2). The resulting output signal (E3)–(E2) was sent to the second channel of the oscilloscope (C2). Needles were inserted into the muscles to a depth of approximately 7 mm. A schematic of the electric circuit is shown in Figure 1.

The RMS (root mean square) voltage value, detected at the second channel of the oscilloscope, was registered every 15 min for 24 h after death. The time series RMS(t) for each specimen were collected and statistically analyzed.

### 2.5. Statistical Tools

To investigate whether the resulting time series associated with the RMS values for each specimen were correlated to each other, we first evaluated the linear correlation coefficient:(1)$$r_{ij}=\frac{\mathrm{cov}\left(T_{i},T_{j}\right)}{\sigma \left(T_{i}\right)\sigma \left(T_{j}\right)}$$
where cov(*T_i_, T_j_*) denotes the covariance between the *RMS* values of specimen *i* and *j*, while *σ*(*T_i_*) is the standard deviation of the RMS values of *i*-th specimen.

Moreover, we looked for the function that best approximated the observed time dependence of the RMS values over the 24 h period. Since, from the biological point of view, it is expected that the RMS value exhibits a maximum after a certain time point from the occurrence of fish death, we produced the following the ansatz:(2)$$RMS(t)=y_{0}+A\exp\left(-\exp\left(-\frac{t-t_{MAX}}{w}\right)-\frac{t-t_{MAX}}{w}+1\right)$$
where *t_MAX_*, *y_0_, A*, and *w* are given constants. This function predicts an initial value:(3)$$RMS(0)=y_{0}+A\exp\left(-\exp\left(\frac{t_{MAX}}{w}\right)+\frac{t_{MAX}}{w}+1\right)$$
followed by a maximum achieved at *t_MAX_*, where the RMS value obtains:(4)$$RMS(t_{MAX})=y_{0}+A$$
and, after that, it exponentially decreases towards the value *y_0_*.

The maximum excursion between the initial value and the value at maximum is given by:(5)$$\Delta V=A\left(1-\exp\left(-\exp\left(\frac{t_{MAX}}{w}\right)+\frac{t_{MAX}}{w}+1\right)\right)$$
whereas the area between the RMS signal and the minimum RMS value, evaluated over the overall time window (24 h), is given by:(6)$$Area=\int_{t=0}^{t=24h}\left\{RMS(t)-\min\left[RMS(t)\right]\right\}\;dt$$

Finally, to test the predictive power of the above fitting, that is, to quantify the goodness-of-fit for linear regression analysis, we also compute the adjusted R-Squared (*adj.R^2^*) value through:(7)$$adj.R^{2}=1-\frac{\left(1-r^{2}\right)\left(N-1\right)}{N-k-1}$$
where *r^2^* is the square of the linear correlation coefficient defined in (1), *N* is the number of points in the dataset, and *k* is the number of independent variables in the model.

To define whether the investigated population can be divided into homogeneous groups with respect to some defined features, a cluster analysis was performed based upon an agglomerative hierarchical clustering method. In the first step, this algorithm requires building up the distance matrix. For this purpose, we computed the standard Euclidean metric among all specimens. Then, the distance among clusters was evaluated using the “average linkage” agglomeration method, where the distance between two clusters is given by the weighted average distance between the elements in two adjacent clusters, with weights proportional to the number of objects inside each cluster. The fusion level of the entities was measured by cophenetic distance (d), representative of the distance between clusters. The objects in the original dataset were then linked together in a graduated hierarchical tree, a so-called dendrogram. Moving up the tree, the objects are combined into branches, which are themselves fused at a higher height (a larger cophenetic distance). The smaller the fusion level, the more similar the objects are. Finally, the desired information on the resulting different clusters was deduced by cutting the dendrogram at a given distance.

All numerical and statistical investigations were carried out by means of MATLAB^®^ (version R2021a).

Finally, a two-way analysis of variance (two-way ANOVA) followed by a Tukey’s test were performed to highlight any significant difference in RMS values between the experimental groups (Group I T 25 °C, Group II T 20 °C, and Group III T 15 °C), based on gender and room temperature conditions. Additionally, a one-way ANOVA was conducted on haematological parameters, to ensure the absence of significant differences between the samples that could alter the accuracy of the measured RMS values. Univariate analyses were performed using PAST V.4.0 software. The *p* value was set at *p* < 0.05.

All data are presented as mean ± SD.

## 3. Results

### 3.1. Study Population and Blood Parameters

A total 18 eighteen fish (14 males and 4 females) were used in this experimental study (body weight: 135.08 ± 13.72 g; length: 23.00 ± 4.50 cm). Fish were allocated into three groups (6 specimens for each group), based on the room temperature set during experimental procedures, as follow:Group I (T_room_ = 25.0 °C): specimens #1–6; Body weight: 134.89 ± 12.59 g; length: 26.50 ± 4.72 cm);Group II (T_room_ = 20.0 °C): specimens #7–12; Body weight: 130.05 ± 11.13 g; length: 21.40 ± 3.51 cm;Group III (T_room_ = 15.0 °C): specimens #13–18; Body weight: 133.34 ± 12.27 g; length: 21.10 ± 3.49 cm).

On clinical examination, all fish were in good health, and none of them showed pathological gross lesions and/or ectoparasites infestation. Neither gross nor histopathological lesions were observed in animal sentinels. Hematological parameters were within the physiological range for the species for all fish [32]: WBC (14.94 ± 1.22 × 10^3^/μL); Rbc (2.74 ± 0.41 × 10^6^/μL); Hgb (6.94 ± 0.93 g/dL); Hct (29.35 ± 4.19%); MCV (117.04 ± 12.11 fL); MCH (27.75 ± 2.29 pg); MCHC (24.16 ± 1.56%); TC (55.51 ± 10.08 × 10^3^/μL). Electrolyte levels were assessed as follows: calcium: 12.6 ± 2.23 mg/dL; sodium: 188.62 ± 12.21 mmol/L; potassium: 4.60 ± 1.69 mmol/L; chloride: 158.96 ± 9.51 mmol/L.

The mean and standard deviation of different hematological parameters and electrolytes levels for fish included in three experimental groups are reported in Table 1. Analysis of variance (one-way ANOVA) did not show any significant difference between groups’ hematological parameters (*p* > 0.05).

### 3.2. Experimental Data: Muscular Electrical Conductivity

The RMS values of the output voltage recorded along a 24 h experimental period for each specimen are reported in Figure 2 and in Appendix A. The results obtained by ANOVA did not show significant differences on RMS value according to the sampled gender (*p* > 0.05), whereas a significant discrepancy on RMS results was observed between the Groups I and III (*p* = 0.001). Although the initial RMS value exhibited some fluctuations among individuals, a similar trend in RMS values during the postmortem interval (24 h) was observed in all animals. In particular, after a short period during which the RMS signal decreased, an increasing trend of the signal was observed for all fish, until it reached a peak/maximum. After that, the RMS value progressively decreased over time. All of the signals recorded at the two channels of the oscilloscope were always in phase, providing a clear indication of the pure resistive nature of the fish tissues. To check the existence of a correlation among the previous time series, we computed the one-by-one correlation coefficients *r_ij_*, Equation (1), and the results are shown in Figure 3. For symmetry, we only represented the lower triangular part. Since *r_ij_* ranged between 0.978 and 0.999, a strong linear correlation was obtained, confirming that the abovementioned time-behavior holds for all specimens.

Then, we explored the possibility of finding the function that best represented the RMS trend during the postmortem interval, exploring the results in Figure 2 by comparing each curve with the best fit arising from the peak functions reported in Equation (2). Results are illustrated in Figure 4, and, for each specimen, we also computed the *adj.R^2^* value (see Figure 5).

The obtained values, *adj.R*^2^ > 0.86, confirmed that the above function can be satisfactorily used to describe the trend of the RMS parameter over the 24 h-period, starting from the fish death. Some trials addressed by means of other peak functions provided lower *adj.R*^2^ values. To better characterize the RMS trend, we investigated the time at which maximum value is reached (*t_MAX_*), and the maximum excursion of the RMS voltage (Δ*V*) (Equation (5)), as well as the area beneath the curve (Equation (6)). Results are presented in Figure 6.

The key result obtained from this figure is that the time when the maximum value is reached strongly depends on the room temperature at which animals are kept. As shown in Figure 6a, the RMS peak value was firstly reached in animals maintained at 25.0 °C (i.e., from about 6 h to less than 8 h). In animals maintained at 20.0 °C, the maximum RMS values were reached from 8 h to 11 h after death, whereas they were reached from 12 h to 14 h in fish maintained at the lowest temperature (i.e., 15.0 °C). The results obtained by ANOVA confirmed the differences observed in RMS peak value, showing a significant *t_MAX_* variability between the experimental groups (*p* < 0.05) (Table 2).

A different consideration can be made about the maximum excursion of the RMS voltage (Δ*V*) and the area depicted by the best fit curve reported in Figure 6b,c. Indeed, except for three isolated cases (specimens #9, #10, and #18), the median values and the standard deviation of the former are M(Δ*V*) = 0.23 V and *σv* = 0.09 V, whereas those of the latter are: M(Area) = 2.98 V·h and σv = 1.87 V·h. Therefore, our data suggest that these parameters are not so crucial in discriminating the postmortem behavior. The possibility of dividing the population into groups was further validated by performing the cluster analysis, based upon an agglomerative hierarchical clustering method. To this aim, the 18 specimens were analyzed according to several features (gender, T_room_, *t_MAX_*, Δ*V*, and Area), as reported in Table 3. Results are shown Figure 7. By cutting the dendrogram at the cophenetic distance *d* = 5, we obtained three different, well separated clusters (note the colors in the figure), which exactly coincide with the three experimental groups corresponding to the room temperature at which fish are kept, and which also identify the different intervals of time in which the maximum of the RMS value is reached.

## 4. Discussion

This study aimed to investigate the electrical conductivity changes of skeletal muscle of sea bass specimens using an oscilloscope coupled with a signal generator, as an innovative technology, in order to find an objective parameter to be used in early postmortem interval estimation. The use of the oscilloscope has been proved as promising technology, with the advantage of being a rapid, non-destructive, and inexpensive method. In particular, RMS was selected out of all measured parameters, and proved promising for PMI estimation as a common trend was observed during the 24 h study period in all animals, even if some variations were initially observed among individuals. In particular, after a short period during which the RMS value decreased, the signal increased to a peak before gradually decreasing over time. The initial decreasing trend of RMS value during the first 0–4 h denotes a reduced conductivity of the cell membranes, and it could be correlated with the onset of the rigor mortis. Similarly, a significant increase in resistance was found during the first hours after haddock fish were sacrificed, as the muscle goes into rigor mortis [31]. The increase in resistance, caused by the reduced conductivity of the cell membranes during the rigor mortis phase, is generally more predominant in muscle tissue with gap junctions [35], and it appears to be related to increased extracellular resistance due to osmotically induced water shifts and consequent cell oedema [36]. Several variables can influence the onset of rigor mortis, including fish species, handling and stress before death, animal health, and environmental conditions [37]. Furthermore, a high variation in the time of rigor mortis onset can be observed in fish even if they are killed under the same conditions, probably due to natural biological variations among individuals [37,38]. Individual biological variations could also explain the initial fluctuations in the RMS values observed in this study. It has been shown that the time from death to the onset of rigor mortis ranges from 45–54 h in rainbow trout [39] to 12–36 h in Atlantic salmon, even if both are kept under the same storage conditions [40]; while the onset of rigor mortis can be as low as 2 h in stressed animals due to the low pre-mortem concentration of glycogen in the muscles [40,41,42], or in fish subjected to electrical stimulation. Above all, the electrical stimulation produces a rapid depletion of high-energy phosphogenes, such as ATP, in muscles, influencing the rapid onset of the rigor mortis, the maximum index, and the resolution phase [40]. The subsequent increasing trend of RMS values observed may reflect the resolution phase of rigor mortis and the loss of selective cell membrane permeability. The rigor development, as well as its duration and resolution, are influenced by many factors, including environmental temperature [43]. In this regard, the duration of rigor is shorter at higher temperatures [43]. Furthermore, electrical conductivity is positively correlated with temperature, with a significant increase of 1.5–5.0% per degree centigrade.

Moreover, in our study, electrical conductivity was correlated with the room temperature, and the time at which the RMS value reached its maximum varied depending on the different room temperatures. Indeed, the linear correlation coefficient between T_room_ and *t_MAX_* gives the value *r* = 0.973, revealing a strong linear positive correlation between these parameters. In particular, the RMS peak value was reached early in animals kept at 25.0 °C, and up to 14 h after death in those kept at the lowest temperature. The loss of selective cell membrane permeability and cell autolysis could be responsible for the increasing trend of RMS values, as a consequence of a reduction in their impedance to electrical stimulation. As PMI increases, the selective membrane permeability is lost, and results in a gradual leak of intracellular fluids out of the cells, as well as passive diffusion of ions in different body compartments. The intra- and extracellular fluids mixing causes an increase in the electrical conductivity of the tissues after death, since both fluids contain abundant electrolytes [30,44]. Furthermore, as metabolic activities at the cellular level continue in the tissues for a variable duration after death, endogenous enzymes promote myofibrillar proteolysis and fat hydrolysis, resulting in disorganization of the muscular structure [45], and decomposition of macromolecules such as protein and DNA into smaller, charged molecules, results in increased conductivity [23], which could explain the progressive increase in RMS value trend observed here. Changes in electrolytic and water contents and cell autolysis occur more rapidly in muscle stored at high temperatures [46], and these changes may explain the differences in the RMS peak values found here. In particular, calcium plays a crucial role in a wide range of physiological processes, including the modulation of excitability and permeability of plasma membranes, muscular contraction, intracellular signal, and nerve signal transduction [47]. During postmortem intervals, changes in calcium concentration are strongly influenced by external temperatures. Generally, calcium concentration increases rapidly, reaching a maximum value 4 h after death at room temperature, then its concentration progressively decreases until it falls below the antemortem level at 24 h [48]. Conversely, variations in serum calcium concentrations occur more slowly at 4 °C, and the maximum concentration is generally reached at 15 h [48]. The progressive decrease of calcium concentration during postmortem could also explain the decreasing trend of RMS values observed in this study. Furthermore, since biological fluids and most tissues show a negative temperature coefficient of resistivity, the progressive cooling of the tissues determines an increase in their electrical resistivity [49]. In our study, the serum concentrations of electrolytes, including calcium, were determined before the fish were euthanized. Unfortunately, no studies are available on the serum concentrations of electrolytes in sea bass specimens, therefore physiological reference ranges for electrolytes are currently lacking. However, since the serum electrolyte levels obtained were similar among all fish enrolled in our study, it can be assumed that these electrolyte values were physiological.

## 5. Conclusions

This study aimed to explore the use of a novel technology to approximate postmortem intervals based on the electrical conductivity changes of skeletal muscle of sea bass specimens during the early postmortem interval. The use of a signal generator/oscilloscope system has proved to be a promising technology in studying dielectric properties of muscle during the early postmortem interval, with the advantage of being a fast, non-destructive, and inexpensive method. The instrumental challenges lie in the measurement setup configuration and data processing. Moreover, while promising, further electrical and theoretical investigations in experimental studies will be needed to standardize and validate this technology before moving from laboratories to field investigations.

## Figures and Tables

**Figure 1 animals-12-01062-f001:**
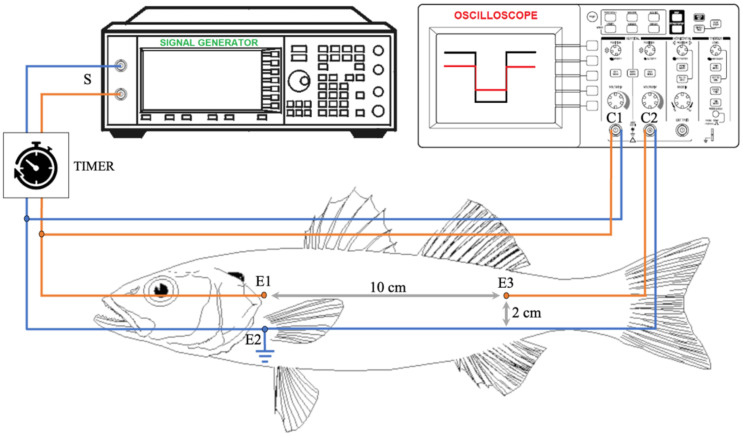
Schematic of the electric system.

**Figure 2 animals-12-01062-f002:**
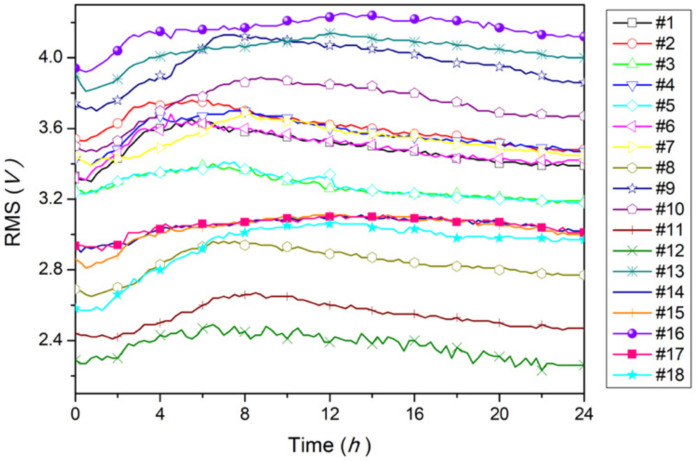
Time evolution of RMS value of the output voltage evaluated over a 24 h period for all 18 sea bass specimens.

**Figure 3 animals-12-01062-f003:**
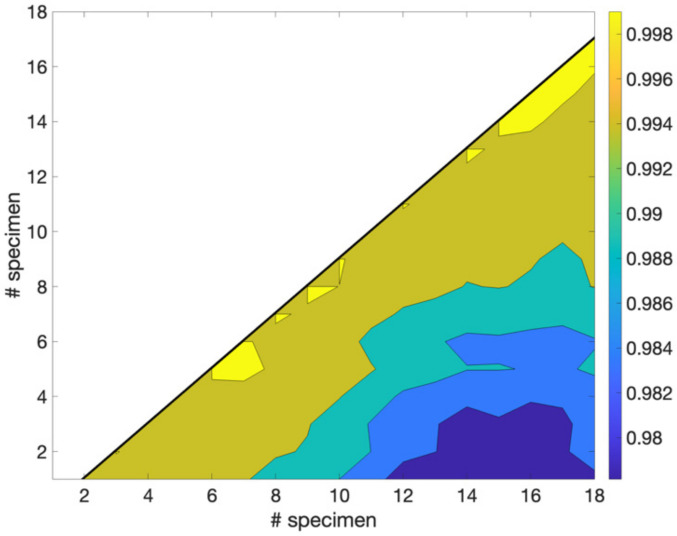
Density plot of the one-to-one linear correlation coefficient *r_ij_*.

**Figure 4 animals-12-01062-f004:**
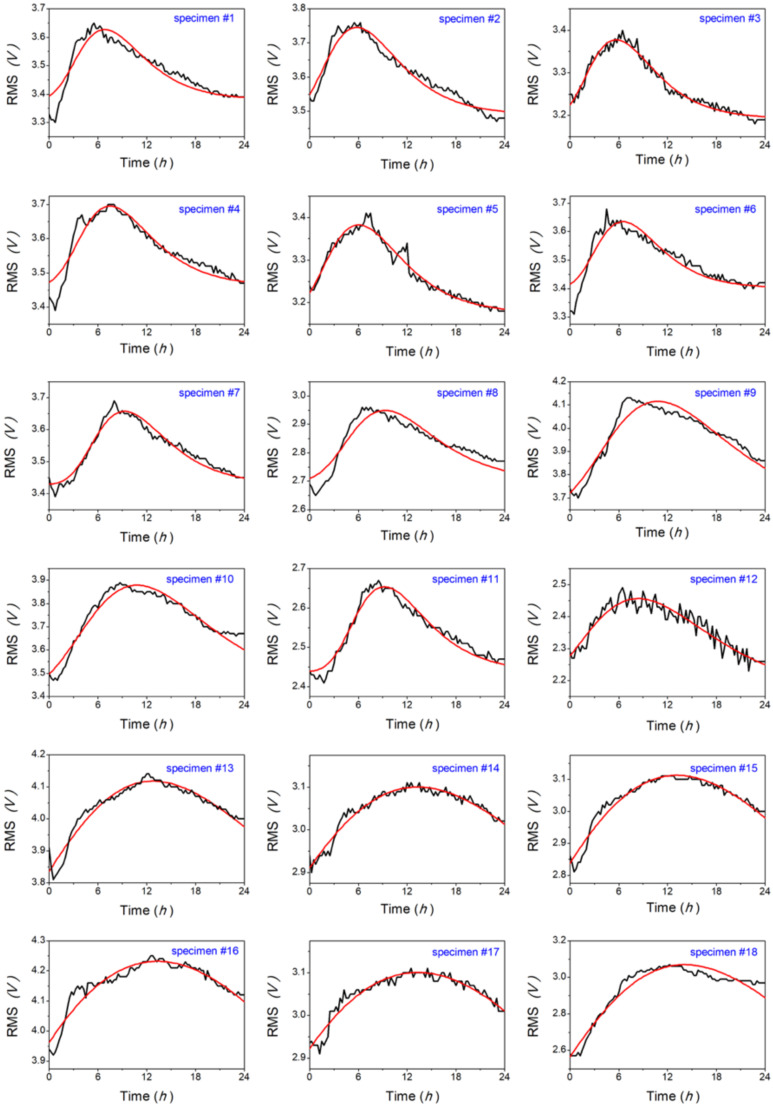
Experimental RMS data (black lines) and best fits obtained using Equation (2) (red lines).

**Figure 5 animals-12-01062-f005:**
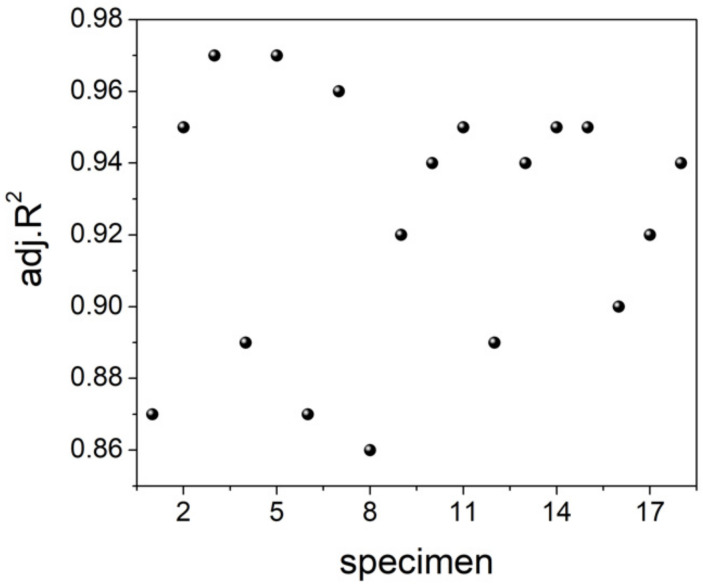
The adjusted R-squared (*adj.R*^2^) values (7) arising from the analysis carried out in Figure 4, evaluated for each specimen.

**Figure 6 animals-12-01062-f006:**
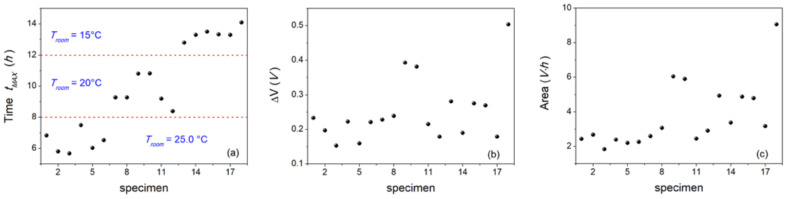
(**a**) Time at which the maximum RMS value is reached (*t_MAX_*); (**b**) the maximum excursion of the RMS value from its initial value (Equation (5)), and (**c**) the area beneath the best fit curve (Equation (6)).

**Figure 7 animals-12-01062-f007:**
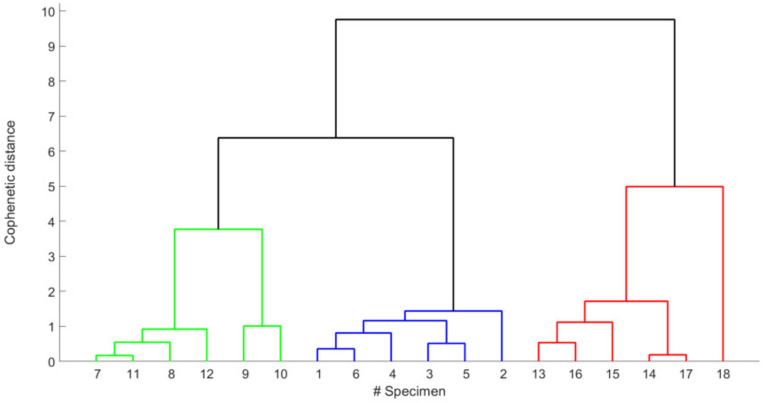
Dendrogram resulting from the agglomerative clustering method.

**Table 1 animals-12-01062-t001:** Mean and standard deviation of hematological parameters and electrolytes measured in fish included in three experimental groups.

	WBC (10^3^/μL)	Rbc (10^6^/μL)	Hgb (g/dL)	Hct (%)	MCV (fL)	MCH (pg)	MCHC (%)	TC (10^3^/μL)	Calcium (mg/dL)	Sodium (mmol/L)	Potassium (mmol/L)	Chloride (mmol/L)
**Mean ± SD** **Group I ** **(25 °C)**	14.57 ± 1.88	2.61 ± 0.53	7.18 ± 1.31	30.22 ± 6.87	116.70 ± 16.99	27.90 ± 3.48	24.04 ± 2.1	56.23 ± 13.54	13.12 ± 1.87	187.67 ± 14.02	4.83 ± 1.97	156.18 ± 11.79
**Mean ± SD** **Group II** **(20 °C)**	14.83 ± 0.75	2.92 ± 0.26	6.77 ± 0.95	28.58 ± 1.49	115.51 ± 10.62	27.60 ± 2.16	24.52 ± 1.50	54.60 ± 9.03	12.17 ± 1.94	191.5 ± 14.36	5.3 ± 1.35	162.35 ± 6.19
**Mean ± SD** **Group III** **(15 °C)**	15.41 ± 0.70	2.68 ± 0.40	6.86 ± 0.46	29.24 ± 2.93	118.91 ± 9.48	27.74 ± 1.02	23.93 ± 1.16	55.69 ± 8.87	12.65 ± 3.01	186.7 ± 9.40	3.67 ± 1.54	158.33 ± 10.31

**Table 2 animals-12-01062-t002:** Results of ANOVA performed on RMS peaks value (*t_MAX_*) reached by the three experimental groups.

Tukey’s Multiple Comparisons Test	Mean Diff.	95.00% CI of Diff.	Significant?	Adjusted *p* Value
*t_MAX_* 25 vs. *t_MAX_* 20	−3.233	−4.710 to −1.757	Yes	0.002
*t_MAX_* 25 vs. *t_MAX_* 15	−6.995	−8.153 to −5.837	Yes	<0.0001
*t_MAX_* 20 vs. *t_MAX_* 15	−3.762	−5.306 to −2.217	Yes	0.0012

**Table 3 animals-12-01062-t003:** Features used in the cluster analysis.

*Specimen*	*gender*	*T_room_* [°C]	*t_MAX_* [h]	Δ*V* [V]	*Area* [V⋅h]
1	M	25	6.84	0.23	2.43
2	F	25	5.80	0.20	2.68
3	M	25	5.67	0.15	1.83
4	M	25	7.49	0.22	2.38
5	M	25	6.03	0.16	2.20
6	M	25	6.52	0.22	2.26
7	M	20	9.27	0.23	2.59
8	M	20	9.28	0.24	3.06
9	M	20	10.80	0.39	6.05
10	F	20	10.82	0.38	5.90
11	M	20	9.19	0.22	2.44
12	M	20	8.39	0.18	2.91
13	M	15	12.80	0.28	4.93
14	M	15	13.30	0.19	3.37
15	F	15	13.50	0.28	4.88
16	M	15	13.32	0.27	4.79
17	M	15	13.30	0.18	3.18
18	F	15	14.10	0.50	9.05

## Data Availability

The data presented in this study are available in Appendix A.

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
