# Peer review of "Postmortem Electrical Conductivity Changes of Dicentrarchus labrax Skeletal Muscle: Root Mean Square (RMS) Parameter in Estimating Time since Death"

_animals, 2022, doi:10.3390/ani12091062_

Round 1
Reviewer 1 Report
In the present manuscript, the authors describe the electric impedance changes in Dicentrarchus labrax muscle after death using an oscilloscope and measuring the Root Mean Square every 15 minutes for 24 hours after death. The data of the present work are of great interest to the scientific community and represent an excellent starting point for the construction of mathematical models to estimate the postmortem interval more accurately in both veterinary and human medicine. These data have a potentially great impact in forensic medicine; however, some minor changes should be made to the manuscript before publication.
Line 125: How much blood was collected from each animal? Could the amount collected affect the measurements? An excessive amount sampled (exsanguination) could alter the electrolyte balance of the tissues.
Line 180: Should the first equation (linear correlation coefficient) be numbered as 1? Could the authors please double-check the numbering of the equations in the section Statistical tools for materials and methods?
Line 199: The version of the MATLAB software used should be indicated in the text.
Line 202: Throughout the study, reference is made to the numbers of individual samples both in the images and in the text. However, it is not clearly stated either in the materials and methods or in section 3.1 how the samples were divided between the groups. I assume that the division is n. 1-6 25°C, n. 7-12 20°C and n. 13-18 15°C. I suggest that the authors clearly indicate the distribution of the samples into the three groups to make the data easier to read.
Line 214: Blood electrolyte values are indicated; however, the authors do not comment on whether these values are within normal ranges for the species. Are there normal ranges of blood electrolytes published in this species? If there are no published ranges in the literature the authors should comment on this result in the discussions in my opinion.
Line 240: What method was used to calculate r2? Please indicate this in the materials and methods.
Also, if the method used includes a p-value, please indicate this in the text.
Furthermore, considering that the paper is also addressed to a reader who has a medical background and does not necessarily have knowledge of statistics, I suggest that the authors introduce the r2 value with a short sentence that makes it clear that it is a correlation coefficient between the curve drawn by the proposed model and the actual observations.
Figure 6a: Are the mean Tmax values of the three groups statistically different? Has a T-test (or the most appropriate test) been performed?
Line 269: In my opinion, it is not possible to conclude that "these parameters are not so crucial in discriminating the postmortem behaviour" based on the data of the present paper. It is possible to assume that the power of the present study is not sufficient to demonstrate a real statistical difference in all examined values. Such differences could be highlighted by a study that includes a larger number of animals. I, therefore, advise the authors to rephrase the sentence by replacing "we can conclude" with "our data suggest".
Author Response
Dear Editor,
Please find our Response to Referees, point-by-point, regarding the revised manuscript ID animals-1676976 entitled "Postmortem electrical conductivity changes of Dicentrarchus labrax skeletal muscle: Root Mean square (RMS) parameter in estimating time since death" by Abbate JM et al.
All corrections have been highlighted in red in the main text and described below line-by-line.
We have revised the paper, considering your suggestions, as follows:
Reviewer(s)' Comments to Author:
Reviewer: 1
In the present manuscript, the authors describe the electric impedance changes in Dicentrarchus labraxmuscle after death using an oscilloscope and measuring the Root Mean Square every 15 minutes for 24 hours after death. The data of the present work are of great interest to the scientific community and represent an excellent starting point for the construction of mathematical models to estimate the postmortem interval more accurately in both veterinary and human medicine. These data have a potentially great impact in forensic medicine; however, some minor changes should be made to the manuscript before publication.
Line 125: How much blood was collected from each animal? Could the amount collected affect the measurements? An excessive amount sampled (exsanguination) could alter the electrolyte balance of the tissues.
Authors’ response: We thank the Reviewer for his/her helpful comment. About 1 ml of blood was sampled for each fish. There is no widely accepted rule for the amount of blood that can be drawn from a live fish, and 0.1%-10% of fish mass blood volumes (BVs) have been suggested. However, BVs do adapt to body size and BV is typically 3% – 4% of body mass in teleost fishes [Lawrence MJ et al., 2020]. Furthermore, caudal puncture is one of the most used techniques, it is rapid and has minimal impacts on fish welfare and therefore on the determination of hematological parameters [Lawrence MJ et al.].
Reference: Lawrence MJ et al., Best practices for non-lethal blood sampling of fish via the caudal vasculature. Journal of Fish Biology. 2020; 97:4-15.
Line 180: Should the first equation (linear correlation coefficient) be numbered as 1? Could the authors please double-check the numbering of the equations in the section Statistical tools for materials and methods?
Authors’ response: The linear correlation coefficient equation (defined in Section 2.5) has been numbered as (1). A check was made for the numbering of the equations.
Line 199: The version of the MATLAB software used should be indicated in the text.
Authors’ response: We used MATLAB ver. R2021a, and this information has been added in the manuscript as suggested.
Line 202: Throughout the study, reference is made to the numbers of individual samples both in the images and in the text. However, it is not clearly stated either in the materials and methods or in section 3.1 how the samples were divided between the groups. I assume that the division is n. 1-6 25°C, n. 7-12 20°C and n. 13-18 15°C. I suggest that the authors clearly indicate the distribution of the samples into the three groups to make the data easier to read.
Authors’ response: The Referee is correct. This information has been clarified in the revised version in Sections 2.3 and 3.1.
Line 214: Blood electrolyte values are indicated; however, the authors do not comment on whether these values are within normal ranges for the species. Are there normal ranges of blood electrolytes published in this species? If there are no published ranges in the literature the authors should comment on this result in the discussions in my opinion.
Authors’ response: We thank the Reviewer for his/her interesting comment and suggestions. Unfortunately, no scientific studies are available relating to the determination of the serum concentration of electrolytes in sea bass specimens, therefore physiological reference ranges for this animal species are currently not present in literature. This comment has been added to the discussion section, as suggested by the Reviewer.
Line 240: What method was used to calculate r2? Please indicate this in the materials and methods.
Also, if the method used includes a p-value, please indicate this in the text.
Furthermore, considering that the paper is also addressed to a reader who has a medical background and does not necessarily have knowledge of statistics, I suggest that the authors introduce the r2 value with a short sentence that makes it clear that it is a correlation coefficient between the curve drawn by the proposed model and the actual observations.
Authors’ response: We thank the Reviewer for his/her useful suggestion. In the Materials and Methods section, the definition of adjusted R2 has been introduced (see eq. (7)), which is the statistical index generally used to evaluate the goodness-of-fit for linear regression analysis. We didn’t use p-value.
Figure 6a: Are the mean Tmax values of the three groups statistically different? Has a T-test (or the most appropriate test) been performed?
Authors’ response: We thank the Reviewer for his/ her comment. To better clarify it, we report here the mean and standard deviation values for each group:
|
Group |
Mean value (h) |
Standard deviation (h) |
|
#1 |
6,39 |
0,70 |
|
#2 |
9,63 |
0,98 |
|
#3 |
13,39 |
0,42 |
In addition, one-way ANOVA followed by a Tukey’s test were performed and all three groups showed significant differences. Results are shown below and reported in the revised version of the manuscript (Table 2).
|
|
|
|
|
|
|
|
Tukey's multiple comparisons test |
Mean Diff, |
95,00% CI of diff, |
Significant? |
Summary |
Adjusted P Value |
|
tmax 25 vs. tmax 20 |
-3.233 |
-4.710 to -1,757 |
Yes |
** |
0.002 |
|
tmax 25 vs. tmax 15 |
-6.995 |
-8.153 to -5,837 |
Yes |
**** |
<0.0001 |
|
tmax 20 vs. tmax 15 |
-3.762 |
-5.306 to -2,217 |
Yes |
** |
0.0012 |
Line 269: In my opinion, it is not possible to conclude that "these parameters are not so crucial in discriminating the postmortem behavior" based on the data of the present paper. It is possible to assume that the power of the present study is not sufficient to demonstrate a real statistical difference in all examined values. Such differences could be highlighted by a study that includes a larger number of animals. I, therefore, advise the authors to rephrase the sentence by replacing "we can conclude" with "our data suggest".
Authors’ response: The manuscript has been corrected according to the Reviewer’s comment.
Reviewer(s)' Comments to Author:
Reviewer: 2
This is an interesting work. The post-mortem electrical conductivity changes were studied on Dicentrarchus labrax skeletal muscle with the aim to stablish or estimate the time since death. The results could be of great value in order to improve this kind of test in field work, being less expensive than molecular estimations.
However, some considerations must be done before article approval. The material and methods and results section need to be improved and some issues clarified. No determination of groups is done on text or figures. No statistical differences are described or showed between different groups (temperatures). For that, to extract a conclusion from these data becomes very difficult. Data process is needed.
Specific comments
Material and methods
Authors included matured fish (male and female) in the study. They found any difference between gender on conductivity? The sea bass is hermaphrodite, why not use immature animals to ensure equal sex?
Authors’ response: We thank the Reviewer for his/her interesting comments. The study population included mainly male sea bass and no differences in electrical conductivity was observed between males and females. In addition, a two-way analysis of variance (Two-Way ANOVA) followed by a Tukey’s test were performed to highlight any significant difference between experimental groups (Group I: T 25 C°, Group II: T 20 C° and Group III: T 15 C°), and no significant differences were found on the RMS value based on to the sample gender (p>0.05). Whereas, a significant discrepancy on RMS results was observed between the groups I and III (p=0.001). This information has been added in the manuscript.
Line 113: After euthanatized, fish were maintaining in water at different temperatures and then, fish were extracted from the water, the electrodes adjusted and then, measurements were done at 20ºC. The electrodes were disposed in the same place? Some tissular damage were noted? Some considerations were done regarding the temperature variations between the water and the measurement place? Was the fish dried before the measurements? Please describe the material and methods carefully and with all details.
Authors’ response: We thank the Reviewer for his/her interesting comments. Live fish were kept in a tank under water-controlled conditions, including water temperature (20-21°C) as specified in Material and methods (section 2.1). Fish were placed in a second tank to perform euthanasia with anesthetic dissolved in the water maintained at the same temperature (20-21°C). immediately afterwards, fish were dried and placed in a room to carry out electrical conductivity measurements under controlled laboratory conditions starting from 20°C as room temperature, as specified in lines 145-146. The electrodes on the animals were always disposed in the same way, with respect to the distance between the electrodes, and no skin damage was observed.
Line 213. The authors only showed the media and standard deviation from the different haematological parameters from all the specimens used in the study which it is practically uninteresting. The media and standard deviation from the different haematological parameters for three experimental groups should be done, as well as the appropriate statistical analysis between them (ANOVA, post-Hoc….) to ensure no differences are found. For example, two females were found in the experiment, any difference can be found.
Authors’ response: We thank the Reviewer for his/her useful comments. The means and standard deviation values from the different hematological parameters for the three experimental groups have been added, accordingly. The descriptive statistic was performed and reported in Table 1. Analysis of variance performed did not show any significant difference between groups’ haematological parameters (p>0.05).
Please, include fish weight and size media for each experimental group. It could be interesting to show if a difference in weight, or fatty acids profile in the muscle, affects conductivity. In this regard, fish size will determine or affect the muscle size (the place in which the electrodes are located) and could be important at time to determine all these parameters (conductivity).
Authors’ response: We thank the Reviewer for his/her comment. Data on body weight and size for fish included in the three experimental groups have been added in the manuscript. As explained in Section 2.4, we have proposed an experimental protocol to make the electrical measurement almost independent of the fish size and weight. Indeed, for each specimen, input and output electrodes were placed in specific points and were maintained at fixed horizontal and vertical distances, thus allowing the inspection of the conductivity properties of the same portion of tissue. In terms of electric circuits, it would correspond to a resistance having the same length and cross-section, but with the only resistivity parameter affected by the postmortem period.
Line 221. You explain results without show them, please, introduce a supplementary file or include in the paper in some way.
Authors’ response: All the results are shown in Figures 2-6. In addition, as suggested, a supplementary file has been added containing all RMS values measured over a 24-hours postmortem period for the 18 sea bass specimens.
Line 232. Figure 2. No determination of groups is done, only the raw data are shown.
Authors’ response: Figure 2 was proposed to show the comparison among the time series of the RMS values associated with all specimens. It was not devoted to determine experimental animal groups. However, since Fig.6 (a) suggests that data can be collected in groups, we also performed a cluster analysis to verify that, taking into account different features (gender, Troom, tMAX, DV and Area), the entire population can be divided into three homogeneous groups, which coincide exactly with the three experimental groups corresponding to the room temperature at which fish were kept, and which also identify the different time intervals at which the maximum of the RMS value is reached. We have added the explanation of the agglomerative cluster method in the materials and methods and results sections, in particular, the new Figure 7, are commented in the “results” section.
Line 242. Figure 4. The units in the y axis are different between all graphics which difficult the comparison among them. Again, no experimental groups are explained (15, 20 and 24ºC), only the specimen number.
Authors’ response: The comparison among all specimens was addressed in Figure 2. In Figure 4, we report the comparison between the experimental data and the best fits. Different units in the y-axis have been introduced just to better appreciate this comparison. Groups discussion are addressed in Figure 6a and 7.
The work comments that specimens 9, 10 and 17 showed discrepancies on Area and delta V values. There is coincidence with female fish?
Authors’ response: Thanks to the Referee for this observation. No, there is no coincidence with female fish, as reported in Table 3. We believe that the discrepancies among specimens #9, #10 and #17 shown in figure 6(b)-(c) cannot be explained by the parameters analyzed here. However, as can also be seen from results of the cluster analysis, they are not so crucial in discriminating the different groups.
Discussion
Line 306: You comment: “…in our study, electrical conductivity was correlated with the room temperature, and the time when the RMS value reached its maximum varied depending on the different room temperatures…”. Where is the model for each temperature (15-20-24ºC)? And the correlation value? How did you statistically demonstrate electrical conductivity maximum depends of temperature?
Authors’ response: We thank the Referee for the suggestion. The model that describes the time dependence of the RMS value is given in eq. (2). A parameter appearing in that model (tMAX) exhibits a clear dependence on room temperature, as argued by the inspection of Fig.6(a) and the results of the cluster analysis (Figure 7), addressed in the revised version of the manuscript. Indeed, as can be seen from the data shown in Table 3, the “gender”, “DV” and “Area” parameters don’t allow to create a sharp separation among the three groups. Moreover, by computing the linear correlation coefficient between “room temperature (Troom)” and “time at which maximum is achieved (tMAX)”, the value r=0.973 is obtained, revealing a strong positive linear correlation between these parameters (this information is added in the revised version). On the other hand, the reduced number of temperatures (just three) does not allow for a more sophisticated investigation of the model that governs such a phenomenon.
What is your proposal for field work? Hypothetical protocol? Has the study considered changes in the environmental temperature? Or the variation of the temperature between the collection of the sample in the field and the electrical conductivity measurement in situ?
Authors’ response: We thank the Reviewer for his/her interesting comments. In this experimental study, the use of an oscilloscope coupled with a signal generator was employed for the first time to evaluate changes in electrical conductivity in skeletal muscles in an experimental animal model. Considering this is a first approach in the field of forensic medicine, this study considers only few variables, such as room temperature (considered as environmental temperature), and evaluate its effect on the measurements. The Authors would demonstrate the potential of this technology in estimating time since death, but of course, numerous other studies will be required under controlled laboratory conditions and introducing numerous other and different variables (animal species; nutrition; habitat, etc.) before being able to transfer this technology into field investigations.

Reviewer 2 Report
This is an interesting work. The post-mortem electrical conductivity changes were studied on Dicentrarchus labrax skeletal muscle with the aim to stablish or estimate the time since death. The results could be of great value in order to improve this kind of test in field work, being less expensive than molecular estimations.
However, some considerations must be done before article approval. The material and methods and results section need to be improved and some issues clarified. No determination of groups is done on text or figures. No statistical differences are described or showed between different groups (temperatures). For that, to extract a conclusion from these data becomes very difficult. Data process is needed.
Specific comments
Material and methods
- Authors included matured fish (male and female) in the study. They found any difference between gender on conductivity? The sea bass is hermaphrodite, why not use immature animals to ensure equal sex?
- Lin 113: After euthanatized, fish were maintaining in water at different temperatures and then, fish were extracted from the water, the electrodes adjusted and then, measurements were done at 20ºC. The electrodes were disposed in the same place? Some tissular damage were noted? Some considerations were done regarding the temperature variations between the water and the measurement place? Was the fish dried before the measurements? Please describe the material and methods carefully and with all details.
- Line 213. The authors only showed the media and standard deviation from the different haematological parameters from all the specimens used in the study which it is practically uninteresting. The media and standard deviation from the different haematological parameters for three experimental groups should be done, as well as the appropriate statistical analysis between them (ANOVA, post-Hoc….) to ensure no differences are found. For example, two females were found in the experiment, any difference can be found.
- Please, include fish weight and size media for each experimental group. It could be interesting to show if a difference in weight, or fatty acids profile in the muscle, affects conductivity. In this regard, fish size will determine or affect the muscle size (the place in which the electrodes are located) and could be important at time to determine all these parameters (conductivity).
- Line 221. You explain results without show them, please, introduce a supplementary file or include in the paper in some way.
- Line 232. Figure 2. No determination of groups is done, only the raw data are shown.
- Line 242. Figure 4. The units in the y axis are different between all graphics which difficult the comparison among them. Again, no experimental groups are explained (15, 20 and 24ºC), only the specimen number.
- The work comments that specimens 9, 10 and 17 showed discrepancies on Area and delta V values. There is coincidence with female fish?
Discussion
- Line 306: You comment: “…in our study, electrical conductivity was correlated with the room temperature, and the time when the RMS value reached its maximum varied depending on the different room temperatures…”. Where is the model for each temperature (15-20-24ºC)? And the correlation value? How did you statistically demonstrate electrical conductivity maximum depends of temperature?-
- What is your proposal for field work? Hypothetical protocol? Has the study considered changes in the environmental temperature? Or the variation of the temperature between the collection of the sample in the field and the electrical conductivity measurement in situ?
Author Response

(The authors gave the same response as above.)
